# Let's Think Frame by Frame with `VIP`: A Video Infilling and Prediction Dataset for Evaluating Video Chain-of-Thought

**Vaishnavi Himakunthala\*, Andy Ouyang\*, Daniel Rose\*, Ryan He\*, Alex Mei,**
**Yujie Lu, Chinmay Sonar, Michael Saxon, William Yang Wang**
University of California, Santa Barbara
{vaishnavi, andyouyang, danielrose, ryanhe, yujielu, saxon}@ucsb.edu
{alexmei, csonar, william}@cs.ucsb.edu

## Abstract

Despite exciting recent results showing vision-language systems' capacity to *reason about images* using natural language, their capacity for *video reasoning* remains underexplored. We motivate framing video reasoning as the sequential understanding of a small number of keyframes, thereby leveraging the power and robustness of vision-language while alleviating the computational complexities of processing videos. To evaluate this novel application, we introduce **VIP**[1], an inference-time challenge dataset designed to explore models' reasoning capabilities through *video chain-of-thought*. Inspired by visually descriptive scene plays, we propose two formats for keyframe description: *unstructured dense captions* and *structured scene descriptions* that identify the focus, action, mood, objects, and setting (**FAMOuS**) of the keyframe. To evaluate video reasoning, we propose two tasks: *Video Infilling* and *Video Prediction*, which test abilities to generate multiple intermediate keyframes and predict future keyframes, respectively. We benchmark GPT-4, GPT-3, and VICUNA on VIP, demonstrate the performance gap in these complex video reasoning tasks, and encourage future work to prioritize language models for efficient and generalized video reasoning.

## 1 Introduction

Constituting 65% of all internet traffic in 2023, videos are an area of huge potential for the next chapter of leveraging artificial intelligence (Fu et al., 2021; Zellers et al., 2021; Fu et al., 2023a). For example, Video Question Answering (Lei et al., 2018) and Video Summarization (Xu et al., 2016) are two existing datasets that empirically evaluate video models. Yet, they do not assess more challenging tasks, such as reasoning through specific relationships between multiple frames. Just

like how humans understand videos by processing frames across time steps, AI's ability to accomplish multi-frame reasoning is a core task for video understanding.

Multi-frame video reasoning is bottlenecked by the sheer computing resources needed to process videos, which typically contain 24 frames per second and can vary widely. However, intelligible videos tend to have little variation from frame to frame. Akin to selecting principle components on the axes containing the highest orthogonal variance, picking a small sample of keyframes can capture much of the video's meaning. Understanding a video via extracted keyframes poses the challenge of multi-hop reasoning, which requires stronger language model capabilities.

To evaluate this challenging multi-hop multi-frame video reasoning, we elicit *video chain-of-thought* (VIDEOCOT). We propose two tasks to test such capabilities in existing models – **Video Infilling** and **Video Prediction**. In the Video Infilling setting, the goal is to predict the masked keyframes' descriptions when given the previous and next keyframes' descriptions in sequence as context, following the spirit of masked language modeling. In the Video Prediction, the goal is to predict the descriptions of the next keyframes in the sequence when given a set of previous keyframes' descriptions, similar to the next token prediction task. These tasks can help analyze video generation and whether video models truly understand the dynamic relations between subsequent video frames, given the variable context gap between keyframes.

To benchmark our proposed Video Infilling and Prediction tasks, we construct a dataset by proposing an automated method to extract keyframes. In addition to the frame itself, we propose two textual representations – *unstructured dense captions* and **FAMOuS** *structured scene descriptions* (Figure 1). The unstructured captions are intended to extract more significant and visually descriptive informa-

---

*Denotes equal contribution.
[1] https://github.com/vaishnaviHimakunthala/VIP

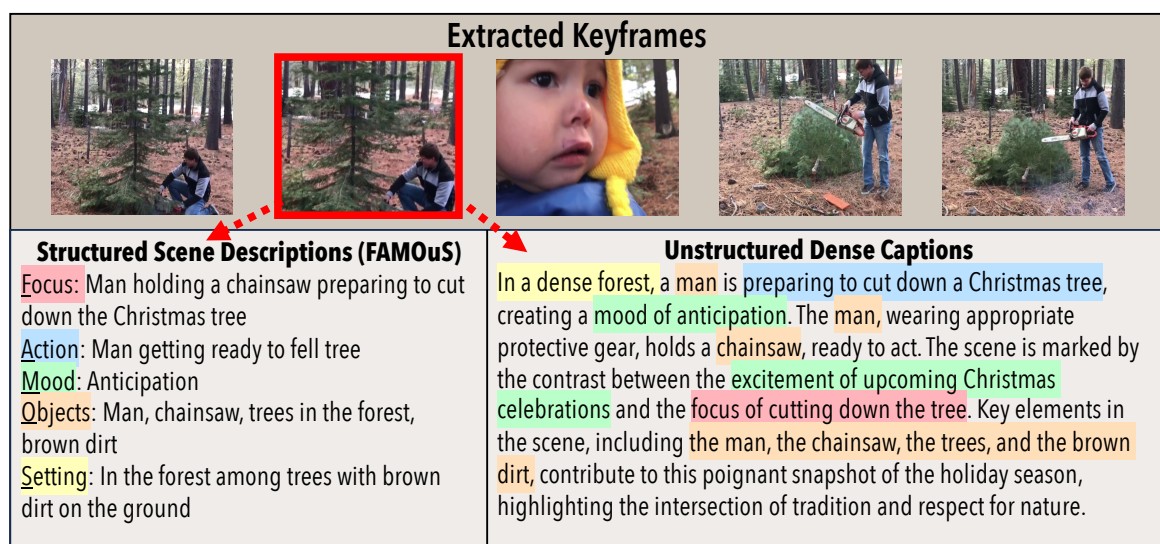

**Extracted Keyframes**

**Structured Scene Descriptions (FAMOuS)**
Focus: Man holding a chainsaw preparing to cut down the Christmas tree
Action: Man getting ready to fell tree
Mood: Anticipation
Objects: Man, chainsaw, trees in the forest, brown dirt
Setting: In the forest among trees with brown dirt on the ground

**Unstructured Dense Captions**
In a dense forest, a man is preparing to cut down a Christmas tree, creating a mood of anticipation. The man, wearing appropriate protective gear, holds a chainsaw, ready to act. The scene is marked by the contrast between the excitement of upcoming Christmas celebrations and the focus of cutting down the tree. Key elements in the scene, including the man, the chainsaw, the trees, and the brown dirt, contribute to this poignant snapshot of the holiday season, highlighting the intersection of tradition and respect for nature.

Figure 1: The Video Infilling and Prediction Dataset consists of two ways to describe keyframes: an **unstructured dense caption** and a **structured scene description** with five components: Focus, Action, Mood, Objects, and Setting (FAMOuS). The unstructured dense captions are highly detailed dense captions that can promote visually descriptive reasoning tasks, while structured scene description provide a concise, visual description of the keyframe that can aid in more focused reasoning tasks.

tion compared to existing captioning systems, a necessary component to provide enough context for more challenging tasks. In addition, we systematically create structured scene descriptions from these unstructured dense captions by extracting the frame's focus, action, mood, objects, and setting using weak human supervision for quality assurance. The FAMOuS categories are inspired by play scripts, which maintain much of the visual detail of the unstructured captions while providing a clear, structured way to reason through visual tasks with high degrees of freedom.

We propose the following contributions:

- We systematically collect an inference-time challenge dataset of keyframes for video reasoning augmented with two textual representations: *unstructured dense captions* for visually-descriptive information and **FAMOuS** *scene descriptions* for structured reasoning.

- We propose the **Video Infilling** and **Video Prediction** tasks to benchmark the *video chain-of-thought* capabilities in existing models.

- We empirically demonstrate that existing models have the potential for multi-hop multi-frame video reasoning tasks but have a significant area for improvement as future work.

## 2   Related Work

**AI Reasoning.**   Large language models (LLMs) demonstrate considerable gains on existing reasoning benchmarks with strategies such as chain-of-thought (CoT) (Wei et al., 2022) and few-shot demonstrations (Brown et al., 2020). Vision-language models (VLM) (Alayrac et al., 2022; Chowdhery et al., 2022; Driess et al., 2023) have furthered LLMs' capabilities by adding the visual modality to perform tasks such as visually-guided text generation (Rose et al., 2023; Zhu et al., 2023), vision question-answering (Wang et al., 2022a; Kim et al., 2021), and image captioning (Li et al., 2022; Liu et al., 2023). We believe the logical next step is to extend these existing models to the video domain. VIP'S annotated collection of extracted keyframes from real-world videos offers a resource to evaluate reasoning abilities within the video domain.

**Datasets for Video Understanding.**   Existing video datasets are often limited by domain specificity or require a supplementary representation (e.g., audio, text) (Lei et al., 2018, 2020a; Tapaswi et al., 2016; Miech et al., 2019; Kim et al., 2017; Mun et al., 2017). These datasets also provide simplifications as textual summaries (Xu et al., 2016; Guadarrama et al., 2013) or a single video frame (Yu et al., 2019; Zeng et al., 2017; Maharaj et al.,

2017), which by themselves can be sufficient to complete the task. While some datasets consider the multi-frame component (Jang et al., 2017; Yi et al., 2020; Mun et al., 2017; Fu et al., 2023b) for higher order complexity, VIP differs in trying to reduce the computational intensity of video reasoning without reducing the task difficulty and generality. VIP is a dataset of real-life videos that spans a breadth of domains and assesses multi-hop, multi-frame video reasoning without requiring significant computation to train on videos.

**Textual Representations of Videos.** Early video models are trained with visual keyframes and textual questions as input and return textual answers as output (Sukhbaatar et al., 2015; Kim et al., 2017; Jang et al., 2017). Then, researchers started to unify the video, keyframe, and text embedding spaces (Miech et al., 2019; Kim et al., 2018; Zellers et al., 2021; Kim et al., 2017; Guadarrama et al., 2013; Bhattacharya et al., 2023). VidIL leverages the contemporary in-context inference paradigm with few-shot demonstrations, including frames, captions, and visual tokens to prompt language models to solve VidL tasks (Lei et al., 2020b; Wang et al., 2022b). In contrast to these existing works, VIP introduces textual representations at the keyframe level and then leverages them to reason about specific video segments using VideoCOT.

# 3 VIP Dataset Construction

To construct the dataset, we first outsource the video corpus to stem from the YouTube-8m dataset (Abu-El-Haija et al., 2016), whose diverse, realistic videos with human-labeled topics align well with our desiderata. To effectively enable multi-frame video reasoning, we downsample visually static categories such as weather, which may not contain much change throughout the video. The weights of each category[2] is described in Figure 2.

Then, to reduce the computational complexity of video processing, we reduce a video into a set of keyframes that seek to capture the video's meaning (§3.1). To accommodate the limitations of existing models, we also generate two forms of visually-descriptive, textual representations– *unstructured dense captions* and *FAMOuS scene descriptions* (§3.2). Figure 3 summarizes this automated pipeline, which reduces the cost ordinarily spent to manually construct such a dataset.

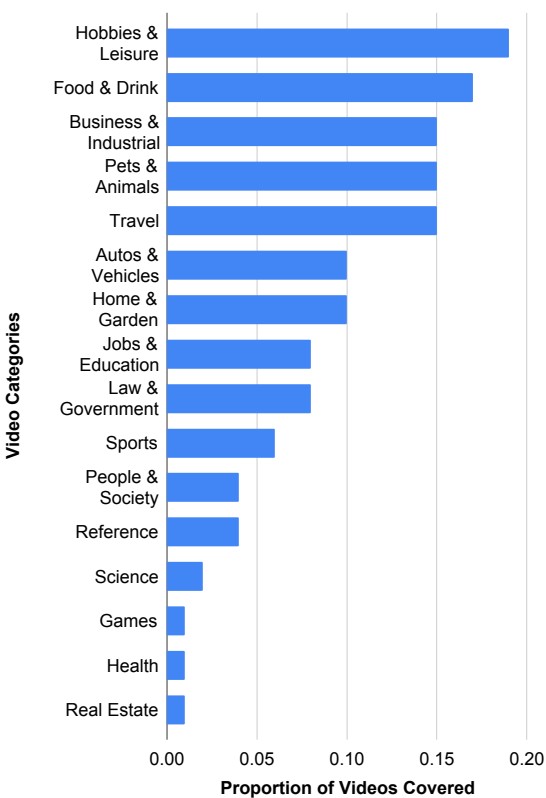

Figure 2: Distribution of VIP's real-world video domains, weighted to emphasize videos containing significant visual change.

## 3.1 Representative Keyframe Selection

**Selecting Video Frames.** The bottleneck to models in the video modality is the computational intensity. We select video frames that best capture the overall video content to mitigate this issue. Instead of training a model to choose a dynamic number of keyframes – which would be computationally expensive – we design an algorithm to prune semantically similar keyframes (Algorithm 1). However, selecting keyframes in this manner comes with the tradeoff that too many frames can introduce redundancy while too few can remove critical context. We choose a large set of candidate keyframes to balance these considerations, which we then dynamically prune. We employ an off-the-shelf keyframe extractor instead of learning a model ourselves. We choose to use KATNA[3] as the baseline keyframe extraction tool as it is open-sourced and easy to onboard. KATNA selects keyframes by leveraging the differences in LUV colorspace, brightness, contrast, blur, and k-means clustering of images.

---

[2]Videos may be in multiple categories.

[3]https://github.com/keplerlab/katna

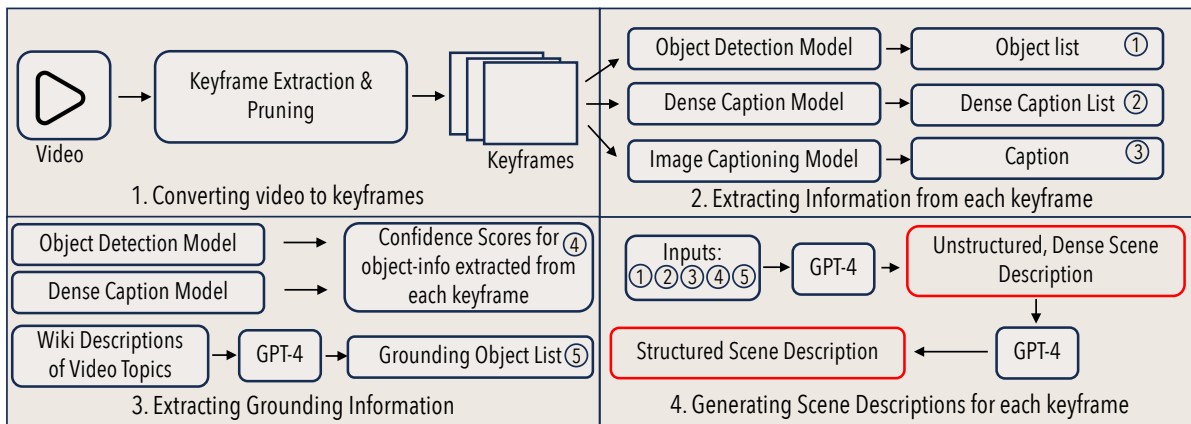

Figure 3: Overview of the pipeline to generate the scene descriptions provided in the VIP Dataset. We first process a video and extract the important frames (§3.1), then generate scene descriptions by extracting visual information from each keyframe, along with grounding information from the video to offset model hallucinations. We then feed in the extracted information into GPT-4 to generate the dense captions and structured scene descriptions. (§3.2).

---

**Algorithm 1:** frame_extract($v, c, f$)

**Data:** video $v$, ints $c, f$ of the candidate and finalized keyframe counts, respectively.

**Result:** List of $f$ finalized keyframes from $v$.

Extract initial frames and embeddings:

1  $k_1, \ldots, k_n \leftarrow Katna(v, c)$
2  $t_1, \ldots, t_n \leftarrow CLIP(Detic(k_1, \ldots, k_n))$
3  $i_1, \ldots, i_n \leftarrow CLIP(k_1, \ldots, k_n)$
4  **while** $len(k) > f$ **do**

    Remove frame with highest adjacent similarity:

5     **for** $j$ in $range(len(k))$ **do**
6        $cos_{text} \leftarrow \cos(t_j, t_{j-1}, t_{j+1})$
7        $cos_{image} \leftarrow \cos(i_j, i_{j-1}, i_{j+1})$
8        $scores[j] \leftarrow mean(cos_{text}, cos_{image})$
9     remove k[s] where s = $\text{argmax}_s(scores[s])$
10 **return** $k_1, \ldots, k_f$

Figure 4: $frame\_extract$ returns a list of $f$ selected keyframes from a video $v$. First, we extract $c$ candidates using Katna. These keyframes are embedded using CLIP in the image space; Detic extracts objects from the keyframes into a textual representation, which are also embedded with CLIP. Then, we iteratively prune the keyframe with the highest cosine similarity with adjacent frames until $f$ keyframes remain.

**Pruning Redundant Frames.** Once baseline candidate keyframes are selected, we prune them by removing low-quality, semantically similar frames. First, we remove blurry keyframes with low Laplacian scores, which indicate the absence of intensity changes. Then, we use object detection models DETIC (Zhou et al., 2022) and GRIT (Wu et al., 2022) to filter keyframes that contain minimal objects, which indicates blurriness as these models are quite sensitive to all background objects.

After removing low-quality frames, we use CLIP

(Radford et al., 2021) to create embeddings for the keyframe image and its list of detected objects and positions from the previous step in the pipeline. This combination helps us compare frames using pixel similarity and object invariance. We take the average cosine similarity score for the keyframe's image and object embeddings compared to the surrounding keyframes and prune the frames with the highest similarity. As people tend to be the primary subject of these videos, we add an additional check only to prune keyframes containing people if either of the surrounding frames also includes people.

## 3.2 Textual Representations of Keyframes

Next, to complement the keyframe images, we construct two textual representations of scenes: an *unstructured, dense caption* that provides visually descriptive insight into the scene; and second, a *FAMOuS scene description* that offers a structured approach to the reasoning process. We first generate the dense caption, which we then use to extract specific information for the structured scene description. These frame descriptions allow for leveraging existing LLM/VLM capabilities for video reasoning and generation.

**Unstructured, Dense Captions.** To create visually descriptive frame descriptions, we first extract three things from each keyframe: a caption, an object list, and a dense caption list. Together, these outputs paint a visual description of the keyframe – the object list and the dense captions describe the focus, objects, and setting, while the caption details the focus, action, and mood. We specifically use

| Dataset | Frame | Structured | Domain | Vid. Len. | Cap. Len. | Test Samples |
|---|---|---|---|---|---|---|
| MSR-VTT (Xu et al., 2016) | ✗ | ✗ | Open | 20.7s | 9.6 | 3K |
| YouCook2 (Zhou et al., 2018) | ✗ | ✗ | Cooking | 5.26m | 8.8 | 2K |
| ActyNet-Cap (Krishna et al., 2017) | ✗ | ✗ | Open | 2m | 13.5 | 5K |
| HowTo100M (Miech et al., 2019) | ✗ | ✗ | Instructional | 18s | 4 | 24K |
| VATEX (Wang et al., 2019) | ✗ | ✗ | Open | 10s | 15.2 | 6K |
| VideoStory (Li et al., 2020) | ✓ | ✗ | Events | 12.6m | 12.1 | 16 |
| WebVid-2M (Bain et al., 2021) | ✗ | ✗ | Open | 18s | 12 | 5K |
| VIP | ✓ | ✓ | Open | 3.6m | 114.2 | 1.5K |

Table 1: Statistics of video datasets: whether they include frame-level (`Frame`) and structured descriptions (`Structured`) as well as the average video token length (`Vid. Len.`), average caption token length (`Cap. Len.`), and number of inference samples (`Test Samples`). VIP is the only video reasoning dataset that has an open video domain *and* frame-level descriptions and provides novel structured frame descriptions.

DETIC (Zhou et al., 2022), a tool that accurately detects objects without much detail, to simply list the frame's objects. To extract more descriptive, object-level detail for high-quality scene descriptions, we use GRIT (Wu et al., 2022), which returns dense captions describing each object. Finally, we obtain the keyframe's overall caption using LLAVA (Liu et al., 2023).

Because these individual models are prone to hallucination (Dai et al., 2023), we ensure the accuracy of our unstructured descriptions by engineering DETIC and GRIT to return confidence scores for each of their outputs. Additionally, we utilize the Wiki descriptions of each video topic in the YouTube-8M dataset to extract a grounding list of baseline objects using GPT-4. Finally, we feed in all of the extracted outputs into GPT-4 to generate the final dense caption.

**FAMOuS Structured Scene Descriptions.** Structure can improve the reasoning ability of a model by providing concrete targets. To provide structure, we take inspiration from scene plays which clearly label and describe the scene. Specifically, we identify and extract the focus, action, mood, objects, and setting from the dense caption using GPT-4, categories which should capture the most important visual information in a concise, structured manner.

### 3.3 Dataset Contributions

The VIP dataset is the first to evaluate multi-hop video reasoning via a video-chain of thought. This novel paradigm promotes efficient and robust video reasoning through automated keyframe extraction (Algorithm 1) over a breadth of domains (Figure 2). Our two textual representations of keyframes (Figure 1) add significantly granularity to videos (with

an average caption length of 114 tokens) compared to traditional video caption datasets (Table 1). This enables reasoning on more specific visual and semantic changes which occur between frames, more closely mimicking how humans process videos by thinking frame by frame.

To ensure the quality of our collected dataset, we verify correctness via crowdsourcing on Amazon Mechanical Turk (Appendix A). Workers are paid to evaluate the quality of structured scene descriptions and edit those of low quality. Unstructured dense captions are corrected using the validated structured scene descriptions with GPT-4 and verified with another round of human evaluation.

## 4 Video Reasoning Tasks

Taking inspiration from existing natural language tasks, we propose two tasks for videos that explore a model's multi-frame reasoning capabilities. The **Video Infilling** task requires models to predict a set of keyframes given the preceding and following frames, akin to masked language modeling for keyframes. **Video Prediction** tasks models to predict the most likely sequence of frames to follow a given series of frames - parallel to the text completion task. Video infilling and prediction of keyframes are two general tasks with several downstream contexts that can benefit from video understanding and completion.

To concretely define the tasks below, we represent the sequence of chronological keyframes as $k_1, \ldots, k_n$, their respective unstructured dense captions as $u_1, \ldots, u_n$, and FAMOuS structured scene descriptions as $s_1, \ldots, s_n$.

### 4.1 Video Infilling Task

Suppose a subsequence of frames $k_i, \ldots, k_j$ is masked. In the video infilling task, the target is for

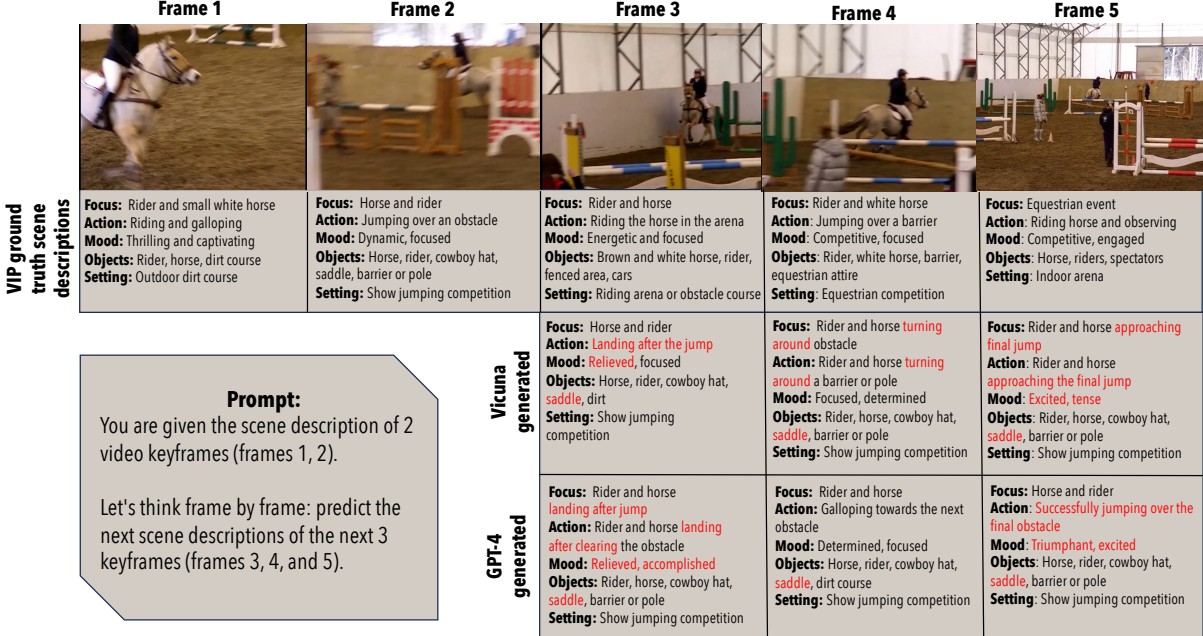

Figure 5: Given a number of context frames, the frame prediction task requires models to predict the following $n$ frames. In this example, we provide two FAMOUS scene descriptions and use VICUNA and GPT-4 to predict the next three frames. Results emphasized in red differ from the ground truth.

a model to learn to reconstruct these masked frames using preceding context frames $k_{i-n}, \ldots, k_{i-1}$ and following context frames $k_{j+1}, \ldots, k_{j+n}$ where $n$ is the number of frames provided as context. Without loss of generality, this task follows for both textual representations using $u$ and $s$ as inputs and outputs instead of $k$. In the multimodal setting, we can use pairs $(k, u)$ or $(k, s)$ as inputs and outputs.

This task requires models to capture a scene's temporal variations and transitions, including changes in visual elements, object positions, and contextual factors. Furthermore, the task's difficulty scales two-fold. First, decreasing the context window $n$ will reduce the ability to leverage hints from surrounding keyframes to infill informatively; combined with the necessity to perform multi-hop reasoning between each pair of frames in sequence, insufficient context could result in training divergence. Second, increasing the number of frames to predict in sequence between $i, j$ also raises similar challenges as too large a gap could add several degrees of freedom, resulting in significant infilling variability. Successfully predicting intermediate keyframes may illuminate models' abilities to reason through the dynamic evolution of scenes and identify critical deltas in videos.

| Metric | Mean $\pm$ STD |
|---|---|
| ROUGE$_L$ | $17.75 \pm 0.43$ |
| BERTSCORE | $18.97 \pm 0.47$ |
| SENTENCEBERT | $53.50 \pm 0.50$ |

Table 2: Average GPT-4 performance across three prompts (Figure 6) on Infilling-1 task reported as mean $\pm$ one standard deviation. The low standard deviation indicates prompt stability.

### 4.2 Video Prediction Task

Suppose we are given a sequence of context frames $k_{i-n}, \ldots, k_i$. In the video prediction task, we aim to predict the $f$ following frames $k_{i+1}, \ldots, k_f$. Without loss of generality, this task follows for the unimodal text and multimodal representations. Much like the infilling task, the difficulty increases by decreasing the context window or increasing the prediction span. Since the prediction task only provides past context, predicting a longer sequence following may be harder as the possibilities increase exponentially.

### 5 Experiments

**Setup.** Although it would be ideal to benchmark multi-modal language models on our proposed tasks, the current pre-trained models (e.g., Open Flamingo and Otter (Awadalla et al., 2023; Li et al.,

| Metric | Model | Infilling-1 | | Infilling-2 | | Prediction-1 | | Prediction-2 | |
|--------|-------|-------------|---|-------------|---|--------------|---|--------------|---|
| | | FAMOuS | Dense | FAMOuS | Dense | FAMOuS | Dense | FAMOuS | Dense |
| ROUGE$_L$ | GPT-4 | 17.34 | 24.92 | 18.44 | 25.25 | 15.52 | 23.31 | 16.66 | 25.22 |
| | GPT-3 | 17.83 | 26.26 | 19.34 | 25.50 | 16.39 | 28.43 | 17.20 | 25.96 |
| | VICUNA | 17.37 | 25.34 | 18.85 | 26.69 | 15.86 | 23.75 | 16.59 | 25.88 |
| BERTSCORE | GPT-4 | 18.61 | 24.81 | 19.66 | 25.67 | 16.24 | 20.57 | 17.24 | 24.79 |
| | GPT-3 | 18.20 | 26.24 | 19.56 | 23.10 | 16.60 | 29.96 | 17.24 | 23.47 |
| | VICUNA | 17.67 | 24.07 | 18.98 | 28.14 | 15.80 | 19.76 | 16.68 | 24.34 |
| SENTENCEBERT | GPT-4 | 53.05 | 58.53 | 53.87 | 58.22 | 50.57 | 53.55 | 51.54 | 57.06 |
| | GPT-3 | 52.95 | 58.54 | 54.57 | 55.69 | 51.04 | 59.83 | 51.81 | 53.99 |
| | VICUNA | 52.19 | 54.66 | 53.33 | 58.80 | 50.40 | 51.86 | 50.96 | 54.86 |

Table 3: Model performance on infilling and prediction tasks when outputting three frames, with the best results underlined.. We vary the number of context frames as indicated by the dash in task name (e.g., Infilling-2 uses two previous and two future context frames to predict three intermediate masked keyframes; Prediction-2, uses two preceding keyframes to predict the subsequent three keyframes). We evaluate models using both FAMOuS structured scene descriptions and unstructured dense captions. Models show weak performance overall.

2023)) are not designed to accommodate multiple image inputs off-the-shelf. As a result, we chose to benchmark the video infilling and prediction tasks as a language task, generating keyframes as represented by dense captions or FAMOuS descriptions. We use GPT-3, GPT-4, and VICUNA[4] as leading models, with in-context inference using one demonstration in both our infilling and prediction tasks. To mitigate hallucination, we leverage greedy decoding. In each task, the goal is to infill or predict *three* intermediate or subsequent keyframes, respectively. Evaluation metrics are computed as the mean of these three generated keyframes compared to the ground truth. Results are reported using one prompt, but a follow-up analysis shows prompt stability through low-variation among other prompts (Table 2).

**Metrics.** We use three standard text comparison metrics: ROUGE$_L$, BERTSCORE (Zhang* et al., 2020), and SENTENCEBERT (Reimers and Gurevych, 2019). ROUGE$_L$ is best suited for tasks aimed to generate text that exactly matches the ground truth. BERTSCORE leverages BERT embeddings, which utilize the surrounding context. SENTENCEBERT is similar to BERTSCORE but computes the similarity of texts using sentence-level embeddings instead of word embeddings. These metrics combined provide initial scope into keyframe generation performance from both the semantic and contextual perspective.

### 5.1 Primary Results

We break down the primary results (Table 3) into four key points.

**Number of Context Frames.** Although the output size is fixed, we investigate how varying the input size affects the complexity of the VIP tasks. Consistent with intuition, we observe higher performance given additional context. However, the performance boost with each additional keyframe is marginal. With *scores for all three metrics significantly lower than other tasks of similar spirit*, it appears that our multi-hop, multi-frame *prediction task is quite challenging using only textual representations for existing state-of-the art language models*. This low baseline performance may overshadow the change in difficulty as a result of varying context frames.

**Dense Captions vs FAMOuS Descriptions.** *Dense captions consistently show stronger performance using our selected metrics than* FAMOuS *descriptions.* As our evaluation metrics emphasize word similarity, they may favor dense captions which contain filler words used to form complete sentences. In the FAMOuS structure, descriptions are broken down by category, which reduces the verbosity, thereby increasing the difficulty for word comparison metrics.

**Infilling vs Prediction Tasks.** We consistently observe that *models have stronger performance on the infilling task compared to the prediction task*. To most fairly compare the two tasks, compare the Infilling-1 task, which aims to predict three intermediate frames given one predecessor and one

---

[4]We use the pre-trained VICUNA-13B checkpoint.

| Metric | Model | FAMOuS | Dense | Focus | Action | Mood | Objects | Setting |
|---|---|---|---|---|---|---|---|---|
| ROUGE$_L$ | GPT-4 | 17.14 | 26.05 | 13.12 | 9.64 | 19.56 | 28.82 | 14.52 |
| | GPT-3 | 17.44 | 29.35 | 14.96 | 10.47 | 19.09 | 27.85 | 14.82 |
| | VICUNA | 17.37 | 26.99 | 15.31 | 10.59 | 19.53 | 28.14 | 13.30 |
| BERTSCORE | GPT-4 | 17.63 | 26.48 | 20.70 | 13.26 | 18.49 | 19.86 | 15.81 |
| | GPT-3 | 17.43 | 30.95 | 22.25 | 14.08 | 16.54 | 19.00 | 15.26 |
| | VICUNA | 17.21 | 26.91 | 21.77 | 13.57 | 18.80 | 18.12 | 13.79 |
| SENTENCEBERT | GPT-4 | 51.87 | 57.89 | 46.83 | 44.48 | 51.34 | 58.84 | 57.88 |
| | GPT-3 | 52.15 | 56.40 | 47.72 | 45.71 | 51.43 | 58.35 | 57.51 |
| | VICUNA | 51.57 | 55.44 | 47.65 | 45.36 | 51.44 | 57.33 | 56.12 |

Table 4: Model performance on the `Prediction-3` task; the two leftmost numerical columns report the aggregate results, while the five rightmost columns report the individual FAMOuS component results. The best results on the component level is underlined for each metric. Language models perform worse predicting `Action` and `Setting` with textual representation inputs, highlighting where video representations could be most beneficial.

successor keyframe, with the `Prediction-2` task, which aims to predict the three keyframes following the two context frames. Aside from a few non-significant outliers, these models perform better across all metrics and both textual representations. This is inline with intuition as *bidirectional context reduces the complexity of the problem.*

**Individual Model Performance.** Across models, *the performance does not follow any obvious trends, which is surprising considering the size difference* in the open-source VICUNA compared to the human-reinforced GPT-3 and GPT-4 models. By metrics, we observe GPT-3 and VICUNA performs slightly better performances on ROGUE$_L$ compared to GPT-4, suggesting exact word consistency may be better in these earlier models. GPT-4's and GPT-3's edge in SENTENCEBERT suggest their generations may align more semantically than an off-the-shelf VICUNA.

## 5.2 FAMOuS Component Analysis

We decompose model performance on FAMOuS structured scene descriptions on a component level (Table 4) for the `Prediction-3` task, which aims to predict the three keyframes following the three input keyframes. Through ROUGE$_L$, we observe models perform significantly better identifying `objects`, compared to the other four components. Comparing BERTSCORE, models appear to semantically align with the ground truth on the `focus` component better and more poorly in understanding of the keyframe's `action`. Finally, the SENTENCEBERT results suggest that models better maintain overall sentence similarity when considering components of the image's environment, such as `mood`, `objects`, and `setting`. These trends

highlight that *reasoning through textual representations for basic components such as keyframe focus and objects is a strength of language models, while reasoning about more dynamic components such as the action necessitates a more intricate understanding of the keyframes* and could benefit from a video representation.

## 5.3 Causal Aspect Analysis

We examine the difference in performance between physical and social causal reasoning (Table 5). A task necessitates physical causal reasoning when the video changes stem from external, tangible forces, like a wave crashing on a beach. Conversely, a task involves social causal reasoning when video changes result from social cues, such as a character becoming joyful during a surprise party. Observation of the results show that *social causal reasoning tasks scored higher on* BERTSCORE *while physical causal reasoning tasks scored higher on* SENTENCEBERT. These results may be an outgrowth of the FAMOuS Component Analysis §5.2, where a consistent character `focus` and `objects` present in many *social* scenarios yield higher token-level similarity with BERTSCORE. By contrast, the consistent environmental qualities like `action` or `mood`– present in many *physical* scenarios– result in a greater SENTENCEBERT score.

## 5.4 Domain Analysis

We also outline the overall results from all experiments corresponding to the different visual domains of our videos in Table 6. Although we found several categories to be noisy due to low sample sizes, *certain categories like* Games *perform well, while others like* Jobs & Education *fall behind.* We hypothesize that the availability of domain-

| Metric | Social | Physical |
|---|---|---|
| ROUGE$_L$ | 18.68 | 17.70 |
| BERTSCORE | 20.81 | 16.61 |
| SENTENCEBERT | 52.70 | 55.44 |

Table 5: GPT-4 performance for videos partitioned by the *physical* (changes due to outside, real forces) and *social* (changes related to social cues) on the Infilling-1 task with structured, FAMOuS descriptions. Significant differences between partitions are underlined.

specific training data as well as intrinsic dimensionality needed to model interactions within these topics jointly contribute to such observations.

### 5.5 Qualitative Observations

Figure 5 provides a visual depiction of the outputs generated by GPT-4 and VICUNA for the prediction task. Inspecting the depicted outputs from both models, it's evident that they *lack some semantic congruence with the ground truth, underscoring the limitations that language model-based approaches face in in video reasoning.* Figure 7 and Figure 5 further demonstrate impressive early performance using video-chain of thought, though the examples' strikingly similar output suggests some overfitting to training data. Despite the observable limitations, it's clear that the language models have a clear baseline video understanding. Still, both the quantitative and qualitative axes highlight that only using unimodal language doesn't generalize their strong language task performance to VIP's video reasoning tasks, which naturally opens several areas for subsequent research threads.

### 6 Future Work

In this paper, we aim to lay the groundwork for exploring the challenging topic of multi-frame, multi-hop reasoning within the existing capabilities of deep learning models. Naturally, this opens several directions for exciting future work.

In the language model space, we benchmark the performance of several leading models using textual representations of keyframes for video-related reasoning following a standard in-context inference procedure with few-shot demonstrations. This invites the opportunity to discover more targeted inference-time techniques using language models or vision-language models to improve the performance of video reasoning tasks beyond a general paradigm. Similarly, additional training-time effort could be worthwhile through fine-tuning or a more traditional train-validate-test paradigm to learn skills beyond the general pre-trained learnset. In this vein, collecting additional data samples could improve the feasibility of these research threads.

Beyond the language modality, bridging the video reasoning task end-to-end with video is a longer-term research direction with immediate benefits in animation. Our paper reduces the video reasoning task into a language task with a textual output. Image synthesis would be an immediate step to reconstruct the keyframe image. Then, video synthesis from a set of images would naturally follow. Finally, unifying these disjoint tasks could benefit from error reduction and improved usability.

As video reasoning is a new space, developing robust evaluation metrics would be a valuable contribution. Some desirable but difficult properties to consider in this area include the ability to capture both the spatial and temporal invariance that could occur through videos, as multiple interchangeable actions are plausible within different areas of the frame and sequences.

Finally, our general video reasoning tasks pose the prospect of efficient transfer learning where improving on such a task could benefit several new applications, similar to the contemporary boom of language technologies.

### 7 Conclusion

We present the inference-time Video Infilling and Prediction dataset to evaluate models' video reasoning abilities by performing a video chain-of-thought on video keyframes. To collect this dataset, we introduce a novel pipeline to systematically extract keyframes and generate corresponding textual representations – unstructured dense captions and structured FAMOuS scene descriptions. We benchmark state-of-the-art language models on VIP tasks and evaluate their ability to generate these textual representations of keyframes through a video chain-of-thought. These models display potential to perform video-related reasoning yet have significant potential to improve. By testing multi-hop, multi-frame reasoning abilities on sparse keyframes, we hope to promote research into developing models that support real-world video understanding and video generation while being resource efficient.

## Limitations

**Model Selection for Benchmarking.** The primary limitation of our work is that current multimodal models do not support multiple image inputs per input query as a trivial use-case. As a result, despite our paper proposing a novel dataset intended for video-related reasoning, we currently only benchmark large language models. We encourage future research to explore the reasoning capabilities for video scene description generation.

**Automation Process.** While our work aims to systematically generate samples to evaluate the video-related reasoning capabilities of existing AI systems, we acknowledge the potential for error when using other AI systems to generate such examples. As a result, we add a layer of human supervision where crowd workers are used to first to classify whether generated scene descriptions are sufficiently correct. Then, we must make use of expert annotators to manually correct the generated scene descriptions that were flagged as poor quality for quality assurances purposes of this dataset.

## Ethics Statement

**Potential for Bias.** We acknowledge the potential for bias in the data collection process and inference task design. We have taken a number of steps to mitigate bias, including ensuring diversity in video content selection, and regularly reviewing and refining the annotation process to minimize any unconscious bias. In addition, we are committed to addressing and correcting any biases that may arise during the evaluation and analysis of VIP model performance. Our dataset is restricted to English captions, thereby containing a bias toward culturally Western scenes. In a multilingual setting differential behaviors between language classes would probably be observed (Saxon and Wang, 2023).

**Crowdsourcing.** Crowdsourcing via the Amazon Mechanical Turk platform was utilized for conducting human evaluation. To ensure reliable results, we restricted participation to workers based in Australia, Canada, New Zealand, the United Kingdom, or the United States, with a minimum HIT approval rating of 98%. Compensation for scene description checking and correction was set at a rate of $15 per hour. Our data collection is classified as an approved exempt protocol from the Institutional Review Board. Details of the interface can be found in Appendix A.

## Acknowledgements

We thank our reviewers for their supportive feedback. This material is based upon work supported in part by the National Science Foundation under Grants #1821415 and #2048122 REU. The authors are solely responsible for the contents of the paper, and the opinions expressed in this publication do not necessarily reflect the official policy or position of associated funding agencies or past or present employers of the authors. The contents of this paper are not intended to provide, and should not be relied upon for, investment advice.

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

## A  Appendix

### A.1  Human Evaluation Interface

We demonstrate the interface of our human evaluation in scene description checking in Figure 9. We employ manual procedures to guarantee the exclusion of personal information and the absence of offensive content during human evaluations.

**Prompt 1 (Used for experimentation):** You are given the scene descriptions of 2 video keyframes (frames 1 and 5). Let's think frame by frame: identify the scene descriptions of the 3 intermediate keyframes that occur between the 2 given scene descriptions (frames 2, 3, and 4). Copy the structure and length exactly.

**Prompt 2:** Your job is to interpolate the scene descriptions for the three intermediate keyframes (frames 2, 3, and 4) using the scene descriptions of the first and fifth keyframes. Ensure that your descriptions match the provided structure and length.

**Prompt 3:** Examine the scene depictions of two video keyframes labeled Scene Description 1 (Frame 1) and Scene Description 5 (Frame 5). Let's break down the progression chronologically: deduce the scene depictions for the three intermediate keyframes (Frames 2, 3, and 4) situated between the two provided scenes. Maintain the identical structure and length precisely.

Figure 6: Input prompt templates used to test for prompt variation (Table 2). To reduce complexity, all subsequent results are reported using Prompt 1, which was designed to align with the widely recognized chain-of-thought prompting approach, adapted for video: "Let's think frame by frame."

You are given the scene descriptions of 2 video keyframes (frames 1 and 5). Let's think frame by frame: identify the scene descriptions of the 3 intermediate keyframes that occur between the 2 given scene descriptions (frames 2, 3, and 4). Copy the structure and length exactly.
Scene Description 1: Focus: Man on fishing boat; Action: Man standing and looking at river; Mood: Focused, calm, enjoyable; Objects: Man, boat, fish food, river, trees, sky; Setting: River in the middle of forest
Scene Description 5: Focus: Man releasing fish into river; Action: Man pushing fish back into water; Mood: Kind, happy, excited; Objects: Man, fish, river, trees, boat, rod; Setting: River surrounded by trees"

Scene Description 2: Focus: Man on boat with fishing rod; Action: Man fishing with rod; Mood: Focused, calm, enjoyable; Objects: Man, fishing rod, boat, river, trees; Setting: River in the forest
Scene Description 3: Focus: Fish in the river; Action: Fish swimming; Mood: Peaceful, calm, still; Objects: Fish, river; Setting: River
Scene Description 4: Focus: Man holding fish on boat; Action: Man taking fish hook out of fish; Mood: Proud, focused; Objects: Man, fish, boat, rod, river; Setting: River

Figure 7: Input prompt with a single in-context demonstration for the `Infilling-1` task.

| Visual Domain | Model | ROUGE$_L$ | | BERTScore | | SentenceBERT | |
|---|---|---|---|---|---|---|---|
| | | **FAMOuS** | **Dense** | **FAMOuS** | **Dense** | **FAMOuS** | **Dense** |
| Jobs & Education | GPT-4 | 0.1570 | 0.2444 | 0.1640 | 0.2337 | 0.4774 | 0.5101 |
| | VICUNA | 0.1662 | 0.2570 | 0.1694 | 0.2599 | 0.4829 | 0.5391 |
| Games | GPT-4 | 0.1793 | 0.2519 | 0.2476 | 0.2619 | 0.5776 | 0.6794 |
| | VICUNA | 0.1821 | 0.2625 | 0.2251 | 0.2892 | 0.5929 | 0.6344 |
| Sports | GPT-4 | 0.1686 | 0.2410 | 0.1940 | 0.2337 | 0.5440 | 0.5808 |
| | VICUNA | 0.1768 | 0.2512 | 0.2000 | 0.2415 | 0.5507 | 0.5546 |
| Pets & Animals | GPT-4 | 0.1644 | 0.2426 | 0.1828 | 0.2366 | 0.5173 | 0.5535 |
| | VICUNA | 0.1642 | 0.2492 | 0.1735 | 0.2309 | 0.5127 | 0.5152 |
| Law & Government | GPT-4 | 0.1484 | 0.2434 | 0.1685 | 0.2223 | 0.5091 | 0.5521 |
| | VICUNA | 0.1501 | 0.2522 | 0.1628 | 0.2285 | 0.5016 | 0.5304 |
| Hobbies & Leisure | GPT-4 | 0.1835 | 0.2590 | 0.1948 | 0.2660 | 0.5511 | 0.6209 |
| | VICUNA | 0.1835 | 0.2640 | 0.1848 | 0.2617 | 0.5434 | 0.5967 |
| Home & Garden | GPT-4 | 0.1656 | 0.2526 | 0.1754 | 0.2430 | 0.5060 | 0.5628 |
| | VICUNA | 0.1601 | 0.2544 | 0.1634 | 0.2282 | 0.4955 | 0.5350 |
| Travel | GPT-4 | 0.1709 | 0.2484 | 0.1855 | 0.2429 | 0.5248 | 0.5539 |
| | VICUNA | 0.1758 | 0.2481 | 0.1816 | 0.2272 | 0.5165 | 0.4978 |
| Food & Drink | GPT-4 | 0.1647 | 0.2482 | 0.1476 | 0.2381 | 0.5171 | 0.5587 |
| | VICUNA | 0.1651 | 0.2509 | 0.1397 | 0.2258 | 0.5048 | 0.5206 |
| Business & Industrial | GPT-4 | 0.1656 | 0.2535 | 0.1716 | 0.2448 | 0.5183 | 0.5820 |
| | VICUNA | 0.1648 | 0.2570 | 0.1654 | 0.2366 | 0.5134 | 0.5501 |
| Autos & Vehicles | GPT-4 | 0.1667 | 0.2476 | 0.1795 | 0.2393 | 0.5396 | 0.5845 |
| | VICUNA | 0.1668 | 0.2509 | 0.1795 | 0.2351 | 0.5311 | 0.5575 |
| People & Society | GPT-4 | 0.1975 | 0.2517 | 0.2242 | 0.2668 | 0.5568 | 0.6402 |
| | VICUNA | 0.2149 | 0.2638 | 0.2263 | 0.2803 | 0.5672 | 0.6304 |
| Reference | GPT-4 | 0.1704 | 0.2590 | 0.1944 | 0.2434 | 0.5147 | 0.5704 |
| | VICUNA | 0.1691 | 0.2550 | 0.1800 | 0.2449 | 0.5072 | 0.5480 |
| Real Estate | GPT-4 | 0.1687 | 0.2590 | 0.1795 | 0.2440 | 0.5270 | 0.6144 |
| | VICUNA | 0.1698 | 0.2647 | 0.1740 | 0.2459 | 0.5172 | 0.6144 |
| Health | GPT-4 | 0.2407 | 0.2442 | 0.2219 | 0.2262 | 0.5141 | 0.5528 |
| | VICUNA | 0.2175 | 0.2551 | 0.1992 | 0.2260 | 0.4894 | 0.5090 |
| Science | GPT-4 | 0.1934 | 0.2436 | 0.2026 | 0.2362 | 0.5492 | 0.5923 |
| | VICUNA | 0.1906 | 0.2448 | 0.1922 | 0.2210 | 0.5434 | 0.5482 |

Table 6: Average model performance across the `Infilling-1`, `Infilling-2`, `Prediction-1`, `Prediction-2`, and `Prediction-3` tasks, partitioned by domain. Categories such as Games, Sports, People & Society, Health, and Science perform better on average, while others such as Jobs & Education, Law & Government, and Food & Drink perform less well. The Best results for each metric and textual representation are underlined.

You are given the scene descriptions of 1 video keyframes (frames 1). Let's think frame by frame: predict the scene descriptions of the next 3 keyframes (frames 2, 3, and 4). Copy the structure and length exactly.
Scene Description 1: Focus: Man on fishing boat; Action: Man standing and looking at river; Mood: Focused, calm, enjoyable; Objects: Man, boat, fish food, river, trees, sky; Setting: River in the middle of forest

Scene Description 2: Focus: Man on boat with fishing rod; Action: Man fishing with rod; Mood: Focused, calm, enjoyable; Objects: Man, fishing rod, boat, river, trees; Setting: River in the forest
Scene Description 3: Focus: Fish in the river; Action: Fish swimming; Mood: Peaceful, calm, still; Objects: Fish, river; Setting: River
Scene Description 4: Focus: Man holding fish on boat; Action: Man taking fish hook out of fish; Mood: Proud, focused; Objects: Man, fish, boat, rod, river; Setting: River

Figure 8: Input prompt with a single in-context demonstration for the `Prediction-1` task.

## Instructions

You will be given a description of an image that is part of a video that describes the image's **focus**, **action**, **mood**, **objects**, and **setting**. Your job is to rate how well the scene description matches the given image.

**Focus:** Reflects the center of activity, attraction, or attention in the image
**Action:** Reflects the primary action occuring in the image
**Mood:** Reflects the feeling or atmosphere set in the image
**Objects:** Reflect all characters and entities that are present in the image
**Setting:** Reflects the place and circumstances at which the image takes place

## Task 1

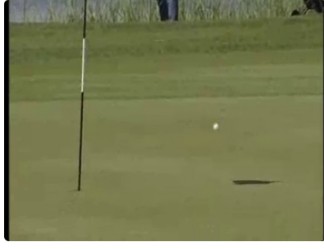

**Focus:** Man playing golf
**Action:** Hitting a golf ball
**Mood:** Pleasant and relaxed
**Objects:** Golf ball, golf club, 14 people, tall grass
**Setting:** Green field, golf course

**The scene description provided may have incorrect or missing information. Rate on a scale of 1-5 on how well the scene description matches the image.**

○ 1 (very poor)   ○ 2 (poor)   ○ 3 (neutral)   ○ 4 (good)   ○ 5 (very good)

Figure 9: `Amazon Mechanical Turk` Platform Interface, where crowdworkers are asked to qualitatively judge the accuracy of a **FAMOuS** scene description.