# OpenReview forum: "Let's Think Frame by Frame with VIP: A Video Infilling and Prediction Dataset for Evaluating Video Chain-of-Thought"
_EMNLP/2023/Conference — EMNLP 2023 Main_

### Official Review · Reviewer_VoFV · 2023-08-02

**Soundness:** 4

**Excitement:**

4: Strong: This paper deepens the understanding of some phenomenon or lowers the barriers to an existing research direction.

**Paper Topic And Main Contributions:**

This paper develop a pipelined approach to extract keyframes and generate a unstructured dense captions scene description and FAMOuS (Focus, Action, Mood, Objects, and Setting) structured scene descriptions.
Moreover, this paper propose the Video Infilling and Prediction inference-time dataset to evaluate video chain-of-thought reasoning with two new tasks in frame generation.

**Questions For The Authors:**

1. Refer to Reasons To Reject.
2. Have you tried to explore the prediction frame description to generate real frame through Stable Diffusion. Some examples would be more interesting.
3. Have your tried different way to select keyframe? E.g, [1]

[1] An Efficient Keyframes Selection Based Framework for Video Captioning, ACL2021

**Reasons To Accept:**

1. The construction of the VIP dataset is valid.
2. The proposed Video Infilling and Prediction inference-time dataset is meaningful to convert video generation into a language-central task, which would be helpful to both video understanding and video generation in the future.
3. Experimental results shows there are still a long way for us in this field.

**Reasons To Reject:**

1. Only an inference-time dataset for VideoCOT is not enough for this task. Although GPT4 and Vicuna have shown great power in language understanding, but the video prediction is still missing during training them, which makes the video understanding and prediction are fur beyond simple COT design. Hope the author can proposed a larger dataset for instruction-tuning.
2. The evaluation of BERT-Score and Rough-L may still pay much attention on token/word-level similarity. May be Sentence-BERT is more applicable for this task.

-------

After rebuttal:
The author have solved my concern.

**Reproducibility:**

4: Could mostly reproduce the results, but there may be some variation because of sample variance or minor variations in their interpretation of the protocol or method.

**Reviewer Confidence:**

4: Quite sure. I tried to check the important points carefully. It's unlikely, though conceivable, that I missed something that should affect my ratings.

---

> ### Author Rebuttal · Authors · 2023-08-29
>
> Thank you Reviewer VoFV for your response! We are happy to hear your appreciation for VIP as well as your shared sentiment on the need for more work in this field. Please consider the following responses to your reasons to reject and your questions as we have conducted supplemental studies based on your insightful recommendations that will go into the camera-ready version.
>
> > 1. Only an inference-time dataset for VideoCOT is not enough for this task. Although GPT4 and Vicuna have shown great power in language understanding, the video prediction is still missing during training them, which makes the video understanding and prediction far beyond simple COT design. Hope the author can propose a larger dataset for instruction-tuning.
>
> We primarily crafted VIP to provide a new evaluation framework and to propose novel tasks for multi-frame video reasoning which is why we focused on making VIP an inference-only dataset. Nonetheless, we see the reviewer’s point on the lack of training on video reasoning tasks in LLMs, therefore we will release an instruction-tuning dataset for the camera ready version of our paper.
>
> > 2. The evaluation of BERT-Score and Rouge-L may still pay much attention to token/word-level similarity. Maybe Sentence-BERT is more applicable for this task.
>
> We had initially omitted metrics like Sentence-BERT because it’s been known to struggle with evaluating fine-grained semantic distinctions, which we had deemed most important for our short FAMOuS descriptions. Still, we are happy to add Sentence-BERT for the camera ready version to benchmark the overall video generation abilities for LLMs.
>
> We ran Sentence-BERT over our full experiments, and the results for infilling 1 (predicting three in-between scene description given one context frame description before and after the scene description to be predicted), infilling 2 (same as infilling 1 but given 2 context frames before and after scene descriptions to be predicted), prediction 1(predicting the next frame description given one context frame description), and prediction 2 (same as prediction 1 but given two context frame descriptions) are in the table below. We found that trends observed with the prior metrics remain consistent with Sentence-BERT, namely on average increasing scores with more provided frames (infilling 1 to infilling 2), and higher scores for the dense caption evaluations. This confirms our initial hypothesis that the performance on VIP tasks depends on the short context phrases in our FAMOuS descriptions, and demonstrates that our observed trends in performance between models and conditions is signal, not noise. Though, Sentence-BERT scores are visibly higher, likely because of a higher level comparison of sentence similarity.
>
>
> **Sentence-BERT scores for Prediction 1 and Prediction 2 tasks**
> | Model   | FAMOuS Prediction 1 | FAMOuS Prediction 2 | Dense Prediction 1 | Dense Prediction 2 |
> |---------|---------------------|---------------------|---------------------|---------------------|
> | GPT-4   | 50.57               | 51.54               | 53.56               | 57.06               |
> | Vicuna  | 50.40               | 50.96               | 51.86               | 54.86               |
>
>
>
> **Sentence-BERT scores for Infilling 1 and Infilling 2 tasks**
> | Model   | FAMOuS Infilling 1 | FAMOuS Infilling 2 | Dense Infilling 1 | Dense Infilling 2 |
> |---------|-------------------|-------------------|-------------------|-------------------|
> | GPT-4   | 53.05             | 53.87             | 58.53             | 58.22             |
> | Vicuna  | 52.19             | 53.33             | 54.67             | 54.86             |
>
>
> > Q2. Have you tried to explore the prediction frame description to generate real frame through Stable Diffusion. Some examples would be more interesting.
>
> This is a point we’ve considered. However, the token limit of Stable Diffusion along with the challenges of semantic generated image evaluation pushed us to focus mainly on textual evaluation. Current image evaluation metrics include using pixel variability and structural similarities which may not accurately capture the semantic meaning of the image. Additionally, image quality is a very subjective topic further complicating image evaluation. For these reasons, we emphasize this for future work.
>
> > Q3. Have you tried different ways to select keyframes?
>
> We thank the reviewer for bringing this up as we put in a lot of thought on our keyframe selection process. We looked into the paper the reviewer suggested, and while we agree strongly with this paper’s goal of reducing redundant information in keyframe extraction, their goal of video captioning is one we differentiate from.  Instead, we focus on multi-frame understanding, which requires maintaining enough frames that still capture the video story.
>
> Furthermore, of this and other available open-source keyframe extraction models, we chose Katna because it is well-documented, high-performing, efficient, and doesn’t require lengthy setup or GPU requirements.  Still, Katna uses state-of-the-art extraction strategies: differences in LUV color space, scores of brightness, contrast, and blur, as well as k-means clustering of image histograms (like in the paper the reviewer suggested). Similarly to the paper, we compare similarity of frames to succinctly capture concepts, though we prune frames to a unique number of keyframes that preserves the videos’ content and length. This is a key difference between our method and what the paper proposes – extracting 3-4 keyframes per video. After much experimentation, we believe that our method effectively selects keyframes that reduce computational power and maintain the integrity of the video for the VIP tasks.

---

### Official Review · Reviewer_6krL · 2023-08-05

**Soundness:** 3

**Excitement:**

4: Strong: This paper deepens the understanding of some phenomenon or lowers the barriers to an existing research direction.

**Paper Topic And Main Contributions:**

Reasoning in text is "reasoning step-by-step". Similar, in video domain, it could be "reasoning frame-by-frame", but only on key frames. In this work, they show how to enable video CoT reasoning w/ existing visual-language models. Specifically, Vision Modules are deployed to extract dense caption descriptions. Language Models are used to perform such few-shot CoT reasoning, with specific two designed tasks (Video Infilling, and Video Prediction) on their *proposed benchmark (VIP, main contribution)*. The experiments with GPT-4 and VICUNA show the difficulties of such LLMs on these video reasoning tasks, asking for more research on video chain-of-thought reasoning w/ current LLMs or VLMs.

**Reasons To Accept:**

1. The idea of multi-modal CoT reasoning (particularly, video modality) is interesting, and the pipeline with unstructured captions looks convincing, by combining the power of several large foundation models.
2. Nice paper structure and organizations.

**Reasons To Reject:**

1. The evaluation of LLMs might be criticized a little bit, since these LLMs are highly sensitive to various prompt formats. Now, one few-shot prompt is used to test their capability in these tasks. Perhaps, it would be good to use some existing techniques to generate more diversified prompts (e.g., zero-shot, or few-shot) with different wordings, e.g., LPAQA w/ paraphrasing for more comprehensive evaluations.
2. In addition to testing on dimensions like "Focus/ Action/ Mood/ Objects/ Setting", it would be good to separate the targeted studies into more fine-grained skills (e.g., visual causal reasoning, understanding dynamics/situations, etc), and visual domains. Therefore, when obtaining the results on your benchmarks, practitioners could get a better intuitive understanding of LLM failures.

**Reproducibility:**

4: Could mostly reproduce the results, but there may be some variation because of sample variance or minor variations in their interpretation of the protocol or method.

**Reviewer Confidence:**

4: Quite sure. I tried to check the important points carefully. It's unlikely, though conceivable, that I missed something that should affect my ratings.

---

> ### Author Rebuttal · Authors · 2023-08-29
>
> Thanks, reviewer 6krL, for your thoughtful response! We are heartened by your appreciation for our pipeline and interest in our novel evaluation direction. Please consider our following responses to your reasons to reject, as we have completed supplemental experiments based on your insightful recommendations that will go into the camera-ready version.
>
> > 1. The evaluation of LLMs might be criticized … [they are] highly sensitive to various prompt formats. … Perhaps, it would be good to use some existing techniques to generate more diversified prompts (e.g., zero-shot, or few-shot) with different wordings, e.g., LPAQA w/ paraphrasing for more comprehensive evaluations.
>
> To validate the stability of our baseline results to prompt variations, we re-ran a subset of our VIP infilling experiments using an additional two new prompts for GPT-4. While our previous prompt used a single few-shot example and prompt phrasing, our two new prompts use two few-shot examples, and two different wordings of the task description generated automatically with GPT-4. Here are the preliminary results:
>
> **Infilling 1 Performance on various paraphrased prompts with Structured, FAMOuS Descriptions**
> | Metric        | Old Prompt Score | New Prompt 1 Score | New Prompt 2 Score |
> | ------------- | ---------------- | ------------------ | ------------------ |
> | Rogue         | 17.34          | 17.70             | 18.20             |
> | Bertscore     | 18.61          | 18.80             | 19.50             |
> | SentenceBERT  | 54.00           | 53.00             | 53.30     |
>
> The FAMOuS results appear to hold across prompt phrasings and one- and two-shot learning. *In the camera ready we will add an additional set of prompt phrasings generated similarly to LPAQA, and we will provide these prompt-specific results for each (model, metric, experiment) combination in an appendix, and averages + std dev for each of these in Tables 2 and 3.* It appears that **overall, the performance of the evaluated baselines holds with simple variations in prompt.**
>
> Here are preliminary results comparing the dense unstructured infilling results from zero-shot and one-shot prompting.
>
> **Infilling 1 Performance on one-shot and zero-shot prompts with Unstructured, Dense captions**
> | Metric        | One-shot score | Zero-shot score |
> | ------------- | ---------------- | ------------------ |
> | Rouge         |         24.92        | 24.72             |
> | Bertscore     |     24.81            | 24.96             |
> | SentenceBERT  |   53.05              | 58.36             |
>
> *The infilling 1 task includes predicting the in-between scene description given one context frame description before and after the scene description to be predicted*
>
> As you can see, the metrics are once again relatively robust to these small variations. Our decision to exclude zero-shot prompting from the original draft is because doing so on the unique FAMOuS schema performs poorly. However, doing so for the dense, unstructured scene description is reasonable. Thus *zero-shot prompting will also be included in our camera ready for the dense infilling and prediction tasks.*
>
> Our selection of prompting techniques is grounded in prior work (Wei et al, 2022). Our use of “let’s think frame by frame”-style prompting is inspired from the well-established “let’s think step-by-step” from Kojima et al (2022).  **We leave deeper exploration of prompting techniques to further optimize performance on this task for future work, as our central goal in this work is producing a benchmark for this novel video understanding task.**
>
> > 2. In addition to … "Focus/ Action/ Mood/ Objects/ Setting", [consider analyzing] fine-grained skills (e.g., visual causal reasoning, understanding dynamics/situations, etc), and visual domains [for] a better intuitive understanding of LLM failures.
>
>  While we have considered breaking our domains down into further categories, ultimately, we aimed to achieve an automatic process such that future work can easily train and evaluate an LLM’s video reasoning capabilities. However, to provide more fine-grained evaluation insights, we have subsampled from our dataset on various categories and provided their results below, which we will add to the camera ready version.
>
> We separated our full studies into the labeled visual domains seen in Figure 4 and found some interesting and consistent trends in performance. The results for the FAMOuS experiment with GPT-4 are highlighted in the table below. We see that categories such as Health & Science, Games, Sports, and Hobbies & Leisure have better performance, while Food & Drink, Law & Government, Jobs & Education don’t perform as well. These results can provide us more insight into which domains future work on LLMs’ video reasoning abilities can focus on.
>
> | Visual Domain      | Rogue   | Bertscore |
> |--------------------|---------|-----------|
> | Jobs & Education   | 15.70   | 16.40     |
> | Games              | 17.93   | 24.76     |
> | Sports             | 16.86   | 19.40     |
> | Pets & Animals     | 16.44   | 18.28     |
> | Law & Government   | 14.84   | 16.85     |
> | Hobbies & Leisure  | 18.35   | 19.48     |
> | Home & Garden      | 16.56   | 17.54     |
> | Travel             | 17.09   | 18.55     |
> | Food & Drink       | 16.47   | 14.76     |
> | Business & Industrial | 16.56 | 17.16     |
> | Autos & Vehicles   | 16.67   | 18.59     |
> | People & Society   | 19.75   | 22.42     |
> | Reference          | 17.04   | 19.44     |
> | Real Estate        | 16.87   | 17.95     |
> | Health             | 24.07   | 22.19     |
> | Science            | 19.34   | 20.26     |
>
> We also labeled a sample of our experiments in our dataset that pertain to two categories of fine-grained skills, namely physical (changes due to outside, real forces) and social (changes related to social cues) causal reasoning. From the results (reported in table below), it's clear that the performance of both types of fine-grained skills is quite similar, showing that there might not be any differences in LLM performance corresponding to these different skills.
>
> | Fine-grained skill | Rouge   | Bertscore | Sentence-BERT |
> |--------------------|---------|-----------|---------------|
> | Social             | 18.68   | 20.81     | 52.70         |
> | Physical           | 17.70   | 16.61     | 55.44         |
>
>
> References:
> - Jason Wei, Xuezhi Wang, Dale Schuurmans, Maarten Bosma, Brian Ichter, Fei Xia, Ed Chi, Quoc Le, Denny Zhou, “Chain-of-Thought Prompting Elicits Reasoning in Large Language Models,” https://arxiv.org/abs/2201.11903 2022
> - Takeshi Kojima, Shixiang (Shane) Gu, Machel Reid, Yutaka Matsuo, Yusuke Iwasawa, "Large language models are zero-shot reasoners." https://proceedings.neurips.cc/paper_files/paper/2022/hash/8bb0d291acd4acf06ef112099c16f326-Abstract-Conference.html 2022

---

### Official Review · Reviewer_JM3S · 2023-08-05

**Soundness:** 4

**Excitement:**

4: Strong: This paper deepens the understanding of some phenomenon or lowers the barriers to an existing research direction.

**Paper Topic And Main Contributions:**

The paper introduces VIP, a novel dataset designed to explore video reasoning capabilities through a video chain-of-thought using keyframes. The authors propose two forms of keyframe descriptions: unstructured dense captions and FAMOuS (Focus, Action, Mood, Objects, and Setting) structured scene descriptions. The dataset includes two tasks: Video Infilling, where models are evaluated on generating intermediate keyframes, and Video Prediction, which tests their ability to predict future keyframes. The authors benchmark GPT4 and VICUNA on VIP and demonstrate the difficulty of complex video reasoning tasks for existing state-of-the-art models.

**Reasons To Accept:**

The paper is well-organized and presents a clear introduction to the problem of video reasoning and the motivation behind VIP.

The paper introduces a pipelined approach to extract the most important frames and generate unstructured scene description. This pipeline can provide more visually-descriptive information enhancing the reasoning capabilities of models in the vision-language domain.

The proposed Video Infilling and Prediction dataset offers a unique perspective on video chain-of-thought reasoning, providing two new tasks for evaluating model performance in generating intermediate and future keyframes. These tasks are valuable additions to the video reasoning research landscape and can lead to advances in video understanding and generation.

The authors perform extensive experiments, benchmarking GPT4 and VICUNA on VIP. The evaluation of existing state-of-the-art models on the proposed tasks highlights the current limitations in complex video reasoning and sets a foundation for future improvements in video understanding models.


**Reasons To Reject:**

none

**Reproducibility:**

4: Could mostly reproduce the results, but there may be some variation because of sample variance or minor variations in their interpretation of the protocol or method.

**Reviewer Confidence:**

2: Willing to defend my evaluation, but it is fairly likely that I missed some details, didn't understand some central points, or can't be sure about the novelty of the work.

---

> ### Author Rebuttal · Authors · 2023-08-29
>
> Thank you reviewer JM3S for your thoughtful and positive response! We are pleased to hear your positive feedback regarding our paper’s organization and the novel pipelined approach for extracting keyframes and generating visually descriptive scene descriptions. Thank you for noting the novelty of our VIP tasks and the uniqueness of our contribution to the video reasoning literature.
>
> You are right that our experimental results show significant room for improvement in video understanding that future systems can aim for. We share your excitement about the future development of video reasoning abilities in LLMs and hope that VIP will inspire advancements in complex video understanding research.

---

### Meta-Review · Area_Chair_M6SZ · 2023-09-21

**Recommendation:** 4

**Metareview:**

This papers presents VIP, an inference-time dataset designed to explore models' reasoning capabilities through video chain-of-thought. To ease the video processing, two forms of textual descriptions are used to describe extracted keyframes, unstructured dense captions and structured scene descriptions that identify the focus, action, mood, objects, and setting. Two tasks are created for the datasets, Video Infilling and Video Prediction. The evaluation is conducted using GPT-4 and Vicuna shows current LLMs still struggle with complex video reasoning. Overall, as pointed out by the reviewers, the paper is well organized with clear motivation, the idea is interesting, the dataset and the pipeline that constructed the dataset could be useful in assisting research in this direction.

---

### Decision · Program_Chairs · 2023-10-07

**Decision:**

Accept-Main

**Comment:**

This papers presents VIP, an inference-time dataset designed to explore models' reasoning capabilities through video chain-of-thought. To ease the video processing, two forms of textual descriptions are used to describe extracted keyframes, unstructured dense captions and structured scene descriptions that identify the focus, action, mood, objects, and setting. Two tasks are created for the datasets, Video Infilling and Video Prediction. The evaluation is conducted using GPT-4 and Vicuna shows current LLMs still struggle with complex video reasoning. Overall, as pointed out by the reviewers, the paper is well organized with clear motivation, the idea is interesting, the dataset and the pipeline that constructed the dataset could be useful in assisting research in this direction.